ecology, evolution, theoretical biology

biodiversity, diversity gradient, stability-time hypothesis, patch-mosaic hypothesis, individual-based simulation

**Author for correspondence:**
Euan N. Furness
e-mail: e.furness19@imperial.ac.uk

# Evolutionary simulations clarify and reconcile biodiversity-disturbance models

Euan N. Furness[1,2], Russell J. Garwood[3,4], Philip D. Mannion[5] and Mark D. Sutton[1]

[1]Department of Earth Sciences and Engineering and [2]Science and Solutions for a Changing Planet DTP, Imperial College London, South Kensington Campus, London SW7 2AZ, UK
[3]Department of Earth and Environmental Sciences, University of Manchester, Manchester M13 9PL, UK
[4]Earth Sciences Department, Natural History Museum, London SW7 5BD, UK
[5]Department of Earth Sciences, University College London, Gower Street, London WC1E 6BT, UK

ENF, 0000-0001-7917-2304; RJG, 0000-0002-2803-9471; PDM, 0000-0002-9361-6941; MDS, 0000-0002-7137-7572

There is significant geographic variation in species richness. However, the nature of the underlying relationships, such as that between species richness and environmental stability, remains unclear. The stability-time hypothesis suggests that environmental instability reduces species richness by suppressing speciation and increasing extinction risk. By contrast, the patch-mosaic hypothesis suggests that small-scale environmental instability can increase species richness by providing a steady supply of non-equilibrium environments. Although these hypotheses are often applied to different time scales, their core mechanisms are in conflict. Reconciling these apparently competing hypotheses is key to understanding how environmental conditions shape the distribution of biodiversity. Here, we use REvoSim, an individual-based, eco-evolutionary system, to model the evolution of sessile organisms in environments with varying magnitudes and scales of environmental instability. We demonstrate that when environments have substantial permanent heterogeneity, a high level of localized environmental instability reduces biodiversity, whereas in environments lacking permanent heterogeneity, high levels of localized instability increase biodiversity. By contrast, broad-scale environmental instability, acting on the same time scale, invariably reduces biodiversity. Our results provide a new view of the biodiversity–disturbance relationship that reconciles contrasting hypotheses within a single model and implies constraints on the environmental conditions under which those hypotheses apply. These constraints can inform attempts to conserve adaptive potential in different environments during the current biodiversity crisis.

## 1. Introduction

Geographic gradients in species richness have been recognized in nature for over two centuries [1], and dozens of hypotheses have been proposed to explain them [2–4]. Despite this, there is still no consensus regarding the causes of these gradients [4,5]. In part, this lack of consensus follows from a lack of clarity surrounding the mechanisms, assumptions and predictions of some proposed models [4]. However, even when predictions for models are formalized, many observations, such as increasing species richness towards the equator, can be used to support numerous hypotheses. These therefore cannot logically be used to confirm any one hypothesis [2,4]. Some workers have suggested that environmental disturbance might influence species richness on an evolutionary time scale [6,7], although the spatial scale on which the evolutionary effects of disturbance has been observed varies between studies: disturbance on large spatial scales [6,8], which is best recorded over geological time scales [9], might have a different impact on species richness to disturbance on a small

spatial scale, where evolutionary time scale effects must be inferred from ecological time scale observations [7,10].

The relationship between environmental stability and species richness over evolutionary time is unclear [11]: two apparently contradictory hypotheses exist. The stability-time hypothesis [2] proposes that environmental stability leads to high species richness as a result of reduced extinction rates [6], increased speciation potential [12–15] or both [16,17]. According to this hypothesis, environmental stability allows organisms to become highly adapted to very narrow environmental niches [12,13,15]. As a result, not only do organisms undergo greater speciation through niche partitioning [18,19], but allopatric speciation [14,15] occurs more frequently as species become limited in their ability to cross environmental barriers. Furthermore, high environmental stability could result in low extinction rates by reducing the rate at which niches are lost due to changing environmental conditions [6,20,21]. By contrast, the patch-mosaic hypothesis [7] suggests that disturbance on small spatial scales can increase species richness over evolutionary time by increasing the ecospace available for niche partitioning in the ecosystem [7,10,22,23].

It is not immediately apparent that these hypotheses are in conflict: in part, because the patch-mosaic hypothesis is often discussed on ecological time scales [24–26], whereas the stability-time hypothesis is typically considered on evolutionary time scales [6,12,15,21]. However, one principle of evolutionary ecology is that the partition of phenomena into evolutionary and ecological time scales should not be an *a priori* assumption [27]. As such, we contend that this distinction in time scales is spurious, and that a conflict does indeed exist. Organisms that are only present in ecosystems because of disturbance on ecological time scales (e.g. ruderal plants [28,29], corals [29]) could not have evolved and cannot persist as species, without a steady supply of disturbed ecosystems over evolutionary time [30–32]. Any hypothesis that ascribes elevated species richness in an ecosystem to a disturbance on ecological time scales (such as the patch-mosaic hypothesis [7]) therefore makes the implicit claim that the repetition of such disturbance over evolutionary time scales is responsible for a portion of the observed species richness. However, this is in conflict with the proposal that ecosystem stability over evolutionary time scales leads to high species richness (i.e. with the stability-time hypothesis).

Eco-evolutionary modelling is emerging as an important tool for understanding the causes of spatial biodiversity gradients: the ability of modelling experiments to independently control variables removes correlations between predictor variables that often hamper observational studies [4]. However, most previous modelling on long evolutionary time scales has been performed at the species level, necessitating that assumptions be made about species-level processes such as speciation and extinction [33,34]. Here, we present a series of experiments investigating the impacts of environmental instability on species richness using REvoSim, a digital eco-evolutionary system that operates at the level of individual organisms and therefore avoids making assumptions about species-level processes. REvoSim models microevolutionary processes such as sexual reproduction, mutation and dispersal within a spatially explicit environment and has been shown to generate, as emergent properties, macroevolutionary phenomena including speciation and adaptation [35]. REvoSim is simplified in a number of ways relative to real ecosystems: (i) it lacks any ecological interactions beyond exploitation competition; (ii) environments only have three axes of variation; (iii) organisms are sessile, facultatively sexual hermaphrodites; and (iv) organisms lack ontogeny or complex behaviour. These simplifications allow computational efficiency and enable the generation of large populations within simulations spanning geological time scales, but also closely reflect the ecology of certain groups (e.g. many plants are sessile, facultatively sexual hermaphrodites). Furthermore, patterns produced by these simulations cannot arise due to processes that do not occur in the model such as interference competition and trophic structure, which are implicated in structuring biodiversity gradients in some hypotheses [36]. Their explicit exclusion from our models enables us to determine if certain patterns can arise without their influence.

In brief, REvoSim models individuals as 64-bit binary genomes that undergo exploitation competition for energy, with fitnesses determined by the interaction between 32-bits of their genome and an environment. Species identity and breed compatibility (i.e. the ability of two organisms to produce offspring through sexual reproduction) are determined by the similarity of genomes between organisms. These organisms occupy single cells in a $100 \times 100$ grid (with a maximum of 100 organisms in each cell) in which cells' environmental conditions are individually specified by three independent variables, visualized as red, blue and green colour intensities. Thus, spatial structure in environments can be achieved by loading images, and the cells' colours can also be varied over time to simulate environmental change (i.e. disturbance). Organisms have a limited lifespan and do not move after their initial dispersal. They primarily reproduce sexually, but asexual reproduction through 'selfing' can also occur, depending on partner availability. Offspring can disperse to nearby cells before competing for energy. The model is described in detail elsewhere [35].

In this study, we conducted three experiments, each of which varied two of the following three variables: (1) pure spatial heterogeneity (PS; figure 1*a*), analogous to permanent environmental heterogeneity within the environment (e.g. altitude differences); (2) pure temporal heterogeneity (PT; figure 1*b*), analogous to large-scale environmental disturbances (e.g. global warming/cooling); and (3) spatio-temporal heterogeneity (ST; figure 1*c*), analogous to small-scale environmental disturbances (e.g. tree falls or mudslides). Simulations were run to equilibrium, and analyses conducted based on the species richness in the equilibrium state. In order to determine whether high species richness is related to rapid speciation or slow extinction, mean species survival durations were also analysed. If the stability-time hypothesis is correct, we would predict that species richness and mean species survival duration will be maximized when both PT and ST are minimized, as both of these factors represent a disturbance in the environment. Alternatively, if the patch-mosaic hypothesis is correct, we would predict that species richness and mean species survival duration will be maximized when ST is maximized, as this latter factor mimics the presence of ephemeral habitat patches, which provide habitats for specially adapted taxa.

## 2. Results

In experiment 1, environments contain only PS and PT. Here the magnitude of PS is positively correlated with species richness ($F = 175.0$, $p < 2.2 \times 10^{-16}$, $n = 180$, d.f. = 1; electronic

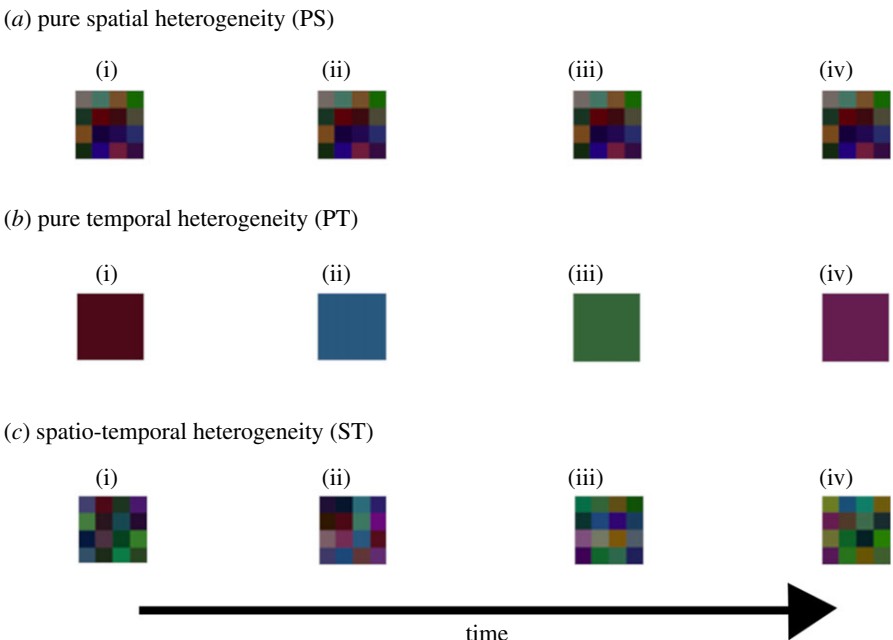

**Figure 1.** Environment components. Four $4 \times 4$ analogues of the $100 \times 100$ grids present in each of the (a) pure spatial heterogeneity (PS), (b) pure temporal heterogeneity (PT) and (c) spatio-temporal heterogeneity (ST) components of the environments. The PS and ST components allow cell colours to differ at any one time in the environment. The PT and ST components allow cell colours to change over time in the environment. This figure was produced using the GIMP graphics program [37]. (Online version in colour.)

supplementary material, dataset S1), and this effect is diminished by an increase in PT ($F = 128.6$, $p < 2.2 \times 10^{-16}$, $n = 180$, d.f. = 1) (figure 2a). In experiment 2, where environments contain only PS and ST, PS magnitude is positively correlated with species richness ($F = 115.0$, $p < 2.2 \times 10^{-16}$, $n = 180$, d.f. = 1; electronic supplementary material, dataset S2), and this effect is generally diminished by an increase in ST ($F = 79.9$, $p < 2.2 \times 10^{-16}$, $n = 180$, d.f. = 1) (figure 2b). However, unlike in experiment 1, the magnitude of the environmental disturbance component (here ST, rather than PT) is positively correlated with species richness when PS is zero ($F = 18.3$, $p = 1.98 \times 10^{-4}$, $n = 30$, d.f. = 1). In experiment 3, environments contain only ST and PT. As in experiment 2, the magnitude of ST is positively correlated with species richness in the absence of PS (PS being absent at all times in experiment 3) ($F = 996.654$, $p < 2.2 \times 10^{-16}$, $n = 1080$, d.f. = 1; electronic supplementary material, dataset S3). This effect is diminished by an increase in PT ($F = 73.873$, $p < 2.2 \times 10^{-16}$, $n = 1080$, d.f. = 1) (figure 2c). Sample sizes are larger in experiment 3 than in either experiment 1 or experiment 2 because maximum species richnesses are much lower, and so noise is more pronounced. Further analysis shows that mean species survival duration (the reciprocal of extinction risk) tracks mean species richness, with both variables responding in the same way to changes in PS, PT and ST (figure 3) (electronic supplementary material, dataset S4).

## 3. Discussion

These findings demonstrate that the stability-time and patch-mosaic hypotheses both affect species richness on an evolutionary time scale. However, the relative dominance of each depends strongly on the degree of PS, and the spatial scale of environmental disturbance. In environments with

PS, our results match the predictions of the stability-time hypothesis [2]: species richness is maximized in the absence of either ST or PT (figure 2). Proponents of the stability-time hypothesis have typically assumed, either implicitly [2] or explicitly [20], that diversification is inevitable given sufficient time. This diversification is typically ascribed to some combination of niche partitioning [13] and allopatric speciation [14,15]. Our results provide empirical support for the assumption that ecosystem stability has no positive effect on species richness in the absence of PS, as even the most stable environments of this type contain only a single niche (i.e. a single environment colour in the simulations), precluding both niche partitioning and allopatric speciation. This also explains why species richness scales with PS: more colours greatly expand the available ecospace and therefore facilitate niche partitioning. The asymptote in species richness is just under 10 000 species in our simulations; this represents saturation because, even with maximal niche partitioning, there can be no more niches than cells in the $100 \times 100$ model environment grid.

When PS is present, species can become specialized in the environments within individual cells. Any amount of disturbance at any spatial scale then reduces species richness, either by changing the conditions within individual cells so as to cause the extinction of specialized species [18,21], or by changing the conditions such that immigrants to cells are no longer less fit than resident individuals, thereby increasing gene flow between cells and inhibiting speciation [19]. On the basis that per-lineage species extinction rates (and at equilibrium, therefore, species turnover rates) are minimized in undisturbed, high PS environments (figure 3), it seems likely that disturbance predominantly reduces species richness through extinction, rather than inhibition of speciation. By contrast, when PS is relatively low, specialization to individual cells is not favoured because cells do

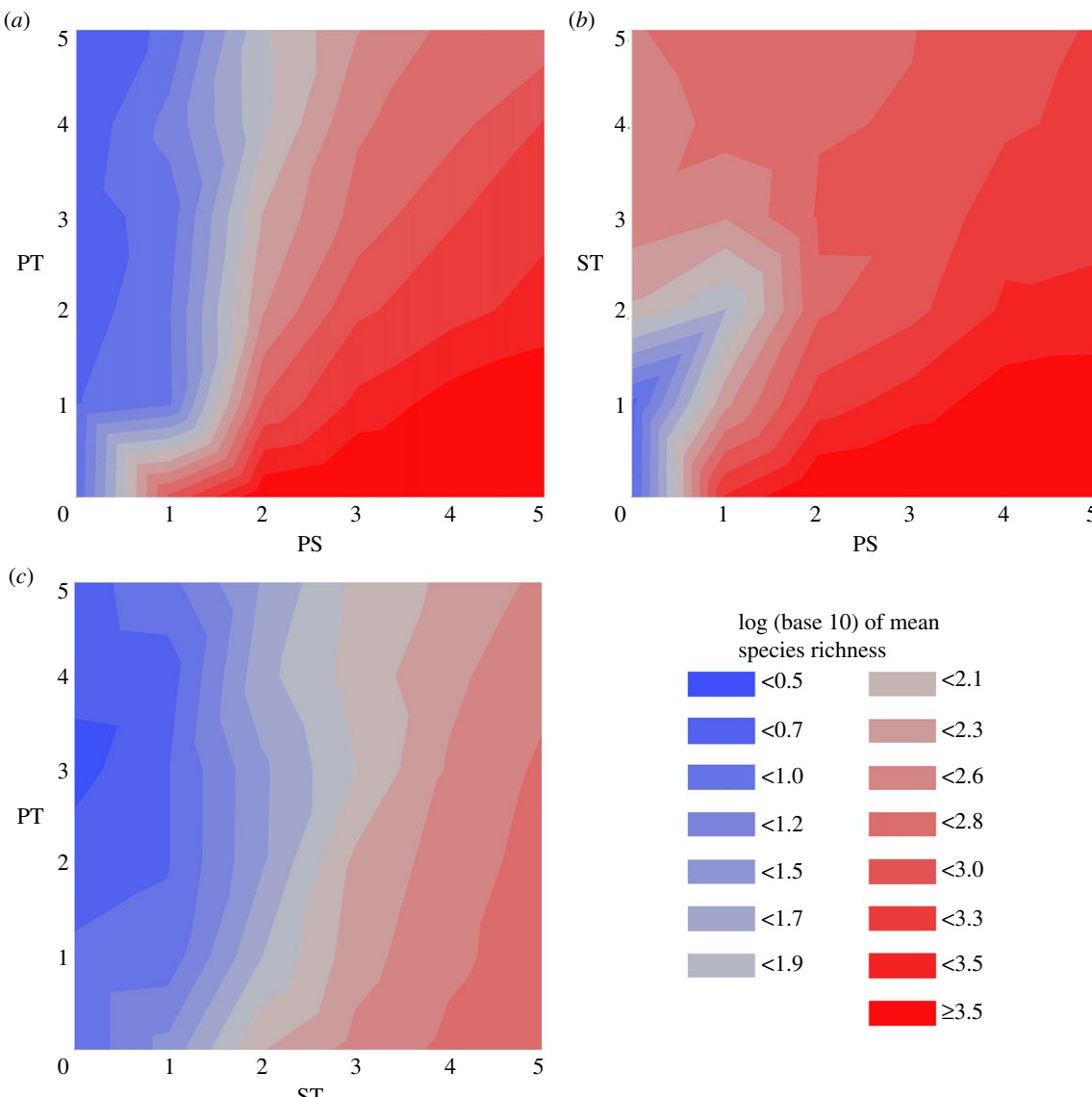

**Figure 2.** Environment impacts on species richness. Surface plots of mean species richness, at the measurement iteration, in simulations run with different magnitudes of two of pure spatial (PS) (*a,b*), pure temporal (PT) (*a,c*) and/or spatio-temporal (ST) (*b,c*) heterogeneity. Each experiment generated one plot: (*a*) experiment 1 (*n* = 180); (*b*) experiment 2 (*n* = 180); (*c*) experiment 3 (*n* = 1080). In all scenarios, PS has a positive impact on species richness and PT has a negative impact on species richness. The impact of ST on species richness is variable; ST has a positive impact when species richness is otherwise low (i.e. when PS is low (part of *b*) or absent (*c*)), but a negative impact when species richness is otherwise high (i.e. when PS is high (part of *b*)). The difference in the impacts of PT and ST on species richness can be seen by comparing the low PS regions (left sides) of (*a*) and (*b*). This figure was produced using JMP PRO [38]. (Online version in colour.)

not have environmental conditions that consistently differ from those in other cells. The negative effects of disturbance on species richness are thus reduced in low PS scenarios. Furthermore, when PS is absent, any small-scale disturbance (i.e. ST) increases the available ecospace by orders of magnitude. This facilitates species specialization not to individual cells, but to particular sets of environmental conditions over a range of cells, although peak species richness here is an order of magnitude below that produced through the stability-time mechanism described above. The mean survival duration of species in these environments is sufficiently high that the species cannot simply be speciating into environments and then going extinct as the environments change (figure 3): instead, they can be inferred to either be continuously adapting to changes in their environments, tracking them through space, or failing to outcompete each other due to the constantly shifting fitness landscape and culling of individuals. This result conforms to the predictions of the patch-mosaic hypothesis [7,10,22], although neutral

processes such as biological cropping [39] may also play a role [40]. By contrast, large-scale disturbance (i.e. PT) invariably reduces both species richness (figure 2) and mean species survival duration (figure 3), regardless of whether that richness is generated as a result of patch-mosaic dynamics (i.e. ST) or PS. Unlike ST, PT does not have the potential to increase the available ecospace in the system and thus facilitate species persistence through niche partitioning, However, it does encourage extinctions and gene flow in the same way as ST.

To interpret our data as support for the patch-mosaic hypothesis, other mechanisms that might explain the positive relationship between small-scale environmental disturbance and species richness must be excluded. In this context, we consider founder effects and neutral (drift) effects below.

One candidate for such an alternative mechanism is founder effects. Environmental disturbance can extirpate species, creating an ecological void [41–43]. Subsequently, the area is likely to be colonized by a relatively small population of

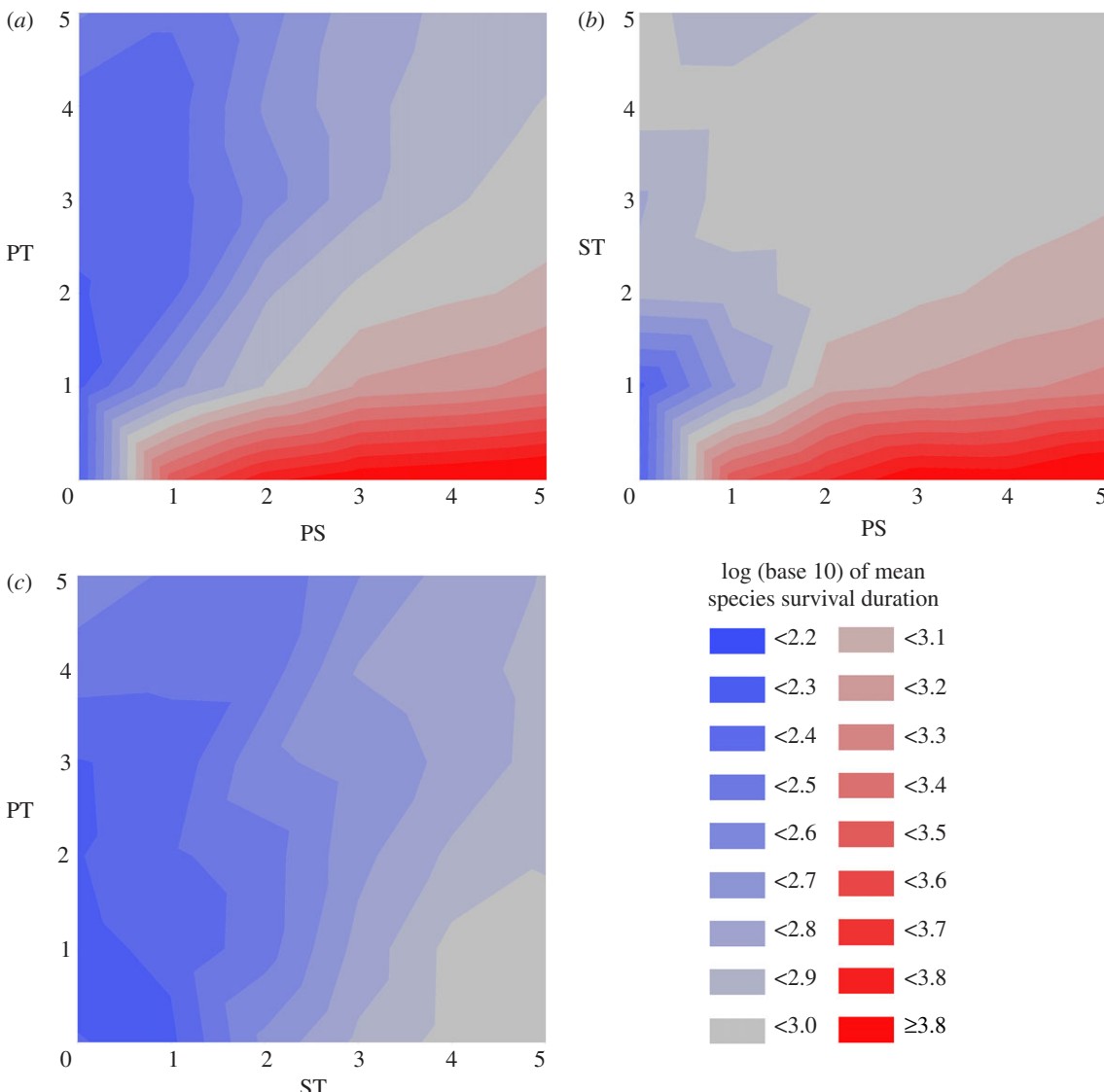

**Figure 3.** Environment impacts on mean species survival duration. Surface plots of mean species survival duration across the measurement period in simulations run with different magnitudes of two of pure spatial (PS) (*a,b*), pure temporal (PT) (*a,c*) and/or spatio-temporal (ST) (*b,c*) heterogeneity. In all scenarios, PS has a positive impact on mean species survival duration and PT has a negative impact. This impact of ST is variable; ST has a positive impact when mean species survival duration is otherwise low (i.e. when PS is low (part of *b*) or absent (*c*)), but a negative impact when mean species survival duration is otherwise high (i.e. when PS is high (part of *b*)). The difference in the impacts of PT and ST on species richness can be seen by comparing the low PS regions (left sides) of (*a*) and (*b*). This figure was produced using JMP PRO [38]. (Online version in colour.)

immigrants, the genetic makeup of which is unlikely to be exactly representative of their parent population, resulting in founder effects [43,44]. In REvoSim, as in the real world, this genetic difference may be substantial [45]. Crucially, however, genetic differentiation as a result of founder effects occurs in REvoSim regardless of the spatial scale of the disturbance. Our observation that PT does not result in increased species richness in our simulations implies that the positive effects of ST on species richness that we observe are not the result of founder effects.

Our experiments could also not have been performed using a neutral (drift) model [40,46]; neutral models can treat disturbance as a process that removes individuals from the system [42], but cannot model disturbance as a change in organism fitnesses because neutral models, by definition, do not calculate such fitnesses. A neutral model could never provide support for the patch-mosaic hypothesis, because this hypothesis requires that disturbance increases species richness by increasing the diversity of niches in an

area [7], a concept that neutral models cannot replicate. Furthermore, in a neutral model, any two of our environments with zero disturbance (i.e. ST = 0 and PT = 0) would be functionally identical and would thus support identical equilibrium species richnesses. This is clearly not the case in our results: disturbance-free environments support vastly more species when they have higher levels of spatial heterogeneity. Similarly, ST and PT heterogeneity will contribute to neutral disturbance processes in a very similar way: by removing individuals from the system. In our results, in the absence of PS, ST correlates positively with species richness, while PT correlates negatively. This difference is not compatible with a neutral process explanation. Given that founder effects and neutral explanations cannot explain our observations, we consider that our results provide support for the patch-mosaic hypothesis.

Our results thus support four key conclusions. First, the explanatory power of the stability-time hypothesis is positively related to the disturbance-independent complexity of the environment (modelled as PS herein). This latter point

was intuited by Dayton & Hessler [39], who suggested that the exceptional biodiversity of the deep-marine environment could not be attributed purely to ecosystem stability. Second, the explanatory power of the patch-mosaic hypothesis is negatively related to the density of disturbance-independent niches in the environment. Third, the spatial scale of environmental disturbance is key: while both small- and large-scale disturbances can produce novel niches, and both will produce vacant ecospace, and therefore encourage speciation through founder effects, only small-scale disturbance increases the total number of niches present in the wider area. As a result, small-scale disturbance can support biodiversity in a way that large-scale disturbance cannot. Finally, if multiple forms of disturbance with different characteristic spatial scales occur within an area, then the smaller scale disturbance can have a positive impact on species richness while the larger scale disturbance has a negative impact. In these scenarios, the patch-mosaic and stability-time hypotheses will be simultaneously applicable. Both hypotheses shape species diversity. We stress that the implication that the stability-time and patch-mosaic hypotheses can occur on the same time scales is an important result, and a conceptual shift from previous work.

The abstract values of PS, ST and PT used herein are not easily comparable with specific metrics of environmental heterogeneity in nature, but our observations nevertheless demonstrate the importance of spatial scale in determining the impact of disturbance on species richness, thus allowing predictions to be made for exemplar ecosystems. For example, many benthic deep-marine environments lack substantial small-scale, disturbance-independent niche diversity [39]. Our results suggest that patch-mosaic effects should predominate in these ecosystems (i.e. ecosystems in which, with a few exceptions [47], almost all niche diversity is generated by localized environmental disturbance [10]). By contrast, the complex physical structure of environments such as mountains provides a relatively high small-scale, disturbance-independent niche diversity. Our results suggest that the stability-time hypothesis will predominate in these environments, and that patch-mosaic dynamics will be less important in maintaining species richness. Regardless of the impact of small-scale disturbance, our results indicate that the relatively low level of large-scale disturbance was probably a major factor in producing the high levels of species richness observed in both the present-day tropics and the deep sea [6,48]: the stability-time hypothesis can hence be considered to apply to both environments, even if biodiversity in the deep sea is elevated as a result of small-scale disturbance. Similarly, cooling events during the Quaternary glaciations represent geologically recent large-scale disturbances at high latitudes [6,49], and our results suggest that the large spatial scale of cooling might be an explanation for the relatively low species richness in affected areas.

Conservation efforts now consider the adaptive potential of species to be an important factor in determining where and how they should be conserved [50], and there is still uncertainty surrounding which areas should be conserved in order to reach international targets [51]. Our conclusions imply that more stable areas should be of greater conservation importance generally, but that less stable areas might be more important in some cases if the instability has a small spatial scale and the environment is otherwise homogeneous (i.e. if ST is high and PS is low).

Evolutionary simulations allow processes that occur on spatial and temporal scales that are not easily observable either in the fossil record, or in the present day to be investigated. Our results demonstrate the contrasting impacts of disturbance on large and small spatial scales, but with the same temporal scale, and provide a framework for the unification of the stability-time and patch-mosaic hypotheses of biodiversity.

# 4. Methods

## (a) Environment generation

Environment grids of $100 \times 100$ cells were generated using a combination of the EnviroGen tool [35] and, for those environments featuring PT, original Python [52] scripts (electronic supplementary material, script S4). Each simulation used 500 environment images that were cycled through at particular rates, with linear colour interpolation between temporally adjacent images. Previous work [35] has shown that REvoSim environments are a viable model for real-world environments, at least to a degree where organisms in REvoSim will evolve to become adapted to a particular environment colour at the cost of adaptations to other colours.

The ST components of the environments (figure 1$c$) were generated using the 'Noise' tab in EnviroGen. The minimum value was invariably set to 0, and the maximum value was set to 51 times the desired ST value (i.e. 0–5) in order to maximize the range of environmental heterogeneity generated. When the desired ST value was 0, images were produced in the GNU Image Manipulation Program (GIMP) [37] instead. The PS components of the environments (figure 1$a$) were generated in the same way but, after the initial environments had been generated, as described for ST, the 'Stack' tab in EnviroGen was used to replace the 500 different noise images with 500 copies of the first image (i.e. a stack of identical images was created). The PT components of the environments (figure 1$b$) were generated in the same way as the ST components, with a different additional step: after the initial environments had been generated, as described for ST, the 'Tile' function (see electronic supplementary material, script S4) was run in Python [52] for each image generated in 'Noise'. For each such image, 'infile' was that image's directory, 'TileWidth' was 1 and 'imSize' was 100. The final output was a series of 500 $100 \times 100$ pixel images, each of a single colour with RGB values between 0 and 51 times the desired PT value (i.e. 0–5). Each output from this function was written to a new image using the 'imwrite' function in the 'imageio' package [53].

The final environments used by REvoSim were then generated by combining either PS and PT environments, PS and ST environments, or ST and PT environments using the 'Comb' tab in EnviroGen, with 'start slice' = 0 and 'Percent influence stack one' = 50 for both the start and end of the combination. This procedure produces environments where the colour of each pixel is the mean of the colours of that pixel in the two environments undergoing combination. This method of combination produces a set of environments that display two different types of heterogeneity simultaneously (equivalent to an environment experiencing, for example, both climate change and landslides). The combination also conserves the relative differences in the magnitude and spatial structure of different environmental heterogeneities. For example, the mean PT in an environment with a PT value of 3 will be three times the magnitude of that in an environment with a PT value of 1, regardless of the other attributes of that environment. Consequently, our method of incorporating disturbance into our simulations creates a scenario where lineages in more highly disturbed environments are more likely than lineages in more stable environments to experience changes in fitness as a result of changes in their environment, as would be expected in nature.

## (b) Running simulations

Simulations were run using settings that have previously been shown to produce biologically realistic outputs [35]. Environment files were selected using the 'Change environment files' tool. Simulations were run for 50 000 iterations each using the 'Run for' tool. Experiment 1 included 180 simulations, each with one of six magnitudes of PS and one of six magnitudes of PT (5 runs of each of 36 possible combinations), where higher magnitudes of a component reflected a greater range of RGB values in that component of the environment. Similarly, 180 simulations were run in experiment 2, each with one of six magnitudes of PS and one of six magnitudes of ST (5 runs of each of 36 possible combinations). A total of 1080 simulations were run in experiment 3, each with one of six magnitudes of ST and one of six magnitudes of PT (30 runs of each of 36 possible combinations).

Different simulations were run to produce logs for the purposes of investigating the impact of PS, PT and ST heterogeneity on mean species survival duration. These simulations were run under identical conditions to the simulations in experiments 1, 2 and 3, with 180 simulations run for each of the three possible pairs of two variables (5 runs of each of 36 possible combinations of those variables in each case).

The details of the REvoSim model, and its implementation, have previously been reported in full by Garwood *et al.* [35]. The simulation settings were as follows: (1) chance of mutation = 10, (2) start age = 15, (3) breed threshold = 500, (4) breed cost = 500, (5) max difference to breed = 3, (6) use max diff to breed = Yes, (7) breed only within species = No, (8) breed mode = sexual, (9) dispersal = 15, (10) nonspatial setting = No, (11) environment refresh rate = 100, (12) environment mode = Loop, (13) interpolate between images = Yes, (14) toroidal environment = No, (15) grid X = 100, (16) grid Y = 100, (17) slots = 100, (18) fitness target = 66, (19) energy input = 2000, (20) settle tolerance = 15, (21) recalculate fitness = No, (22) phylogeny settings = basic, (23) refresh/polling rate = 50, (24) logging: population/environment = None, (25) logging: to text file(s) = write log files, (26) exclude species without descendants = No, (27) minimum species size = 0, (28) don't update gui on refresh/poll = No.

## (c) Extracting data

The output of REvoSim consists of a text file to which a block of text containing the following is added for every extant species every $n$ iterations (where $n$ is the 'Refresh / polling rate'): (i) the unique species ID; (ii) the time in the simulation at which that species was first recorded; (iii) the ID of the species' parent species; (iv) the current population size of the species and (v) the current modal binary genome for the species.

Species richness for each simulation run was extracted from this log using an original Python [52] function (electronic supplementary material, script S1), with 'InputFile' as the directory path to the log file, 'checkpoints' as the number of iterations after which the species richness was to be recorded (45 000), and 'extraRecord' as 0.

Mean species survival durations were determined using electronic supplementary material, script S2. For each simulation log, this script output the sum of the species richnesses in all iterations between 30 000 and 45 000 inclusive, as well as the sum of the species extinctions and originations in this same period. From this, the mean per-species extinction rate was calculated as the number of extinctions divided by the number of species, and the mean survival duration of a species was calculated as

the reciprocal of this rate. This survival duration was then multiplied by 50 to convert from logging iterations to model iterations.

This calculation of mean species survival duration is liable to by heavily influenced by small, ephemeral species [54]. Such species are likely to be superficially very similar to their parents and may not even be detected in the real world. To remove the influence of these species, logs used in the calculation of mean species survival durations were pre-treated to remove species that were present for fewer than 3 logging iterations in the simulations. This pre-treatment used electronic supplementary material, script S3, where 'inputstring' was the path to the log being treated, 'outfile' was the name given to the treated log, 'minIts' was 100 and 'minSize' was 1 000 000.

## (d) Statistical analyses

Statistical analyses of the impacts of PS, ST and PT on species richness were conducted in R [55]. In experiment 1, the impacts of PS and PT on species richness, under all possible combinations of PS and PT, were determined using a standard linear model with interaction terms:

$$Richness = a * PS * PT + b * PS + c * PT + d.$$

In experiment 2, the impacts of PS and ST on species richness, under all possible combinations of PS and ST, were determined using a standard linear model with interaction terms:

$$Richness = a * PS * ST + b * PS + c * ST + d.$$

When considering only those simulations where PS = 0, the impact was determined using a standard linear model:

$$Richness = a * ST + b.$$

In experiment 3, the impacts of ST and PT on species richness, under all possible combinations of ST and PT, were determined using a standard linear model with interaction terms:

$$Richness = a * ST * PT + b * ST + c * PT + d.$$

These models determined the values of the coefficients $a$, $b$, $c$ and $d$ (where applicable) that: (i) produced a mean residual error of zero among all points and (ii) minimized the total absolute magnitude of residual error. The $F$-values reported are the test statistics from ANOVA linear regressions, and $p$-values are derived from these same tests.

Data accessibility. Data used in figure production and statistical analyses are available in the electronic supplementary information (electronic supplementary material, datasets S1–S4). The most recent version of the REvoSim eco-evolutionary system and EnviroGen tool can be downloaded from https://github.com/palaeoware/revosim. The versions used in this paper are archived with Zenodo: doi:10.5281/zenodo.2531611.

Authors' contributions. E.N.F. and M.D.S. conceived the study, with suggestions from R.J.G. E.N.F. ran the simulations, analysed the data, and produced the figures. All authors interpreted the data, contributed to the writing of the manuscript and read and approved the final manuscript.

Competing interests. The authors declare no competing interests.

Funding. E.N.F. was supported by funding from the Natural Environment Research Council (NERC award NE/S007415/1). R.J.G. was supported by NERC award NE/T000813/1. P.D.M. was supported by a Royal Society University Research Fellowship (UF160216).

Acknowledgements. We thank two anonymous reviewers and the editor for comments that improved this contribution.

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
