## [Peer Review File · Proceedings of the Royal Society B: Biological Sciences]

Review History

RSPB-2020-2093.R0 (Original submission)

Review form: Reviewer 1

Recommendation

Reject – article is not of sufficient interest (we will consider a transfer to another journal)

Scientific importance: Is the manuscript an original and important contribution to its field?

Marginal

General interest: Is the paper of sufficient general interest?

Marginal

Quality of the paper: Is the overall quality of the paper suitable?

Acceptable

Is the length of the paper justified?

Yes

Should the paper be seen by a specialist statistical reviewer?

No

Do you have any concerns about statistical analyses in this paper? If so, please specify them explicitly in your report.

No

It is a condition of publication that authors make their supporting data, code and materials available - either as supplementary material or hosted in an external repository. Please rate, if applicable, the supporting data on the following criteria.

Is it accessible?

Yes

Is it clear?

Yes

Is it adequate?

Yes

Do you have any ethical concerns with this paper?

No

Comments to the Author

This paper describes a model run of the REvoSim model, where the 100x100 grid is allowed to vary spatially and temporally in different combinations. The overall goal of the paper is to contrast two hypotheses for local diversity, the "patch mosaic" and "climate stability" hypotheses.

The methods as such seem soundly performed, but I'm not convinced that the design of the study actually tests the two hypotheses, nor am I convinced that this model run in general represents a sizable advance in ecological or evolutionary understanding.

First, the two hypotheses seem to not really be alternatives - patch mosaic is to my mind a more ecological hypothesis at recent time scales, whereas I think of environmental stability as a deeper-time evolutionary hypothesis. There is thus no reason to see them as competing. I'm also unsure how the process of combining e.g. pure spatial with spatiotemporal variation - the key approach in the paper - tests these hypotheses. The process of combining itself - which takes the centroid of two numbers in 3d space - does not have a clear ecological / evolutionary interpretation, at least not one explained in the paper.

So in conclusion I find it hard to imagine ecologists reading this paper and significantly changing their view or understanding of these processes. It could possibly still be interesting to publish it in a journal like PLOS ONE.

A very minor detail: it seems a little strange to call a standard linear model with an interaction a "multiplicative" model.

Review form: Reviewer 2

Recommendation

Major revision is needed (please make suggestions in comments)

Scientific importance: Is the manuscript an original and important contribution to its field?

Good

General interest: Is the paper of sufficient general interest?

Excellent

Quality of the paper: Is the overall quality of the paper suitable?

Good

Is the length of the paper justified?

Yes

Should the paper be seen by a specialist statistical reviewer?

Yes

Do you have any concerns about statistical analyses in this paper? If so, please specify them explicitly in your report.

No

It is a condition of publication that authors make their supporting data, code and materials available - either as supplementary material or hosted in an external repository. Please rate, if applicable, the supporting data on the following criteria.

Is it accessible?

Yes

Is it clear?

Yes

Is it adequate?

Yes

Do you have any ethical concerns with this paper?

No

Comments to the Author

This paper presents an interesting model analysis exploring the potential contribution of disturbance and environmental heterogeneity to biodiversity. The topic is clearly potentially interesting to a broad audience. The paper is generally well written, but it could be made more accessible to a broad audience.

Major comments: I would like to see a clearer communication of the results. The paper's key findings are portrayed in Fig 2, but to the untrained eye the effects of disturbance etc are not easy to ascertain from these plots - which look superficially similar to one another. The Fig legend is also inadequate to explain how these plots communicate the key message around disturbance and biodiversity.

There are numerous ecological parameters included in the models but not extensively explored (competition, dispersal, fecundity, etc). I wonder if comparable results may be obtainable, for instance, from ecologically neutral modelling approaches (e.g. de Aguiar et al 2009 Nature)? To what extent does scale/size of the disturbance interact with the dispersal ability of the organisms?

The MS could also do a clearer job integrating its findings with real-world disturbance scenarios, with more discussion of disturbance-prone systems (compared to the current discussion focus primarily on relatively undisturbed environments such as the deep sea and 'tropics'). E.g. there are some nice examples of ecosystem disturbance and associated biodiversity impacts emerging from studies of tectonically-disturbed intertidal systems (e.g. Parvizi et al 2020 Proc R Soc B).

Minor -

The MS is generally well written, although the first paragraph of the intro becomes almost circular/contorted. Essentially saying: "There are many hypotheses proposed to explain latitudinal biodiversity gradients (sentence 1) because latitudinal diversity gradients can be explained by several different processes (sentence 3)" - sentence 3 seems rather redundant.

Decision letter (RSPB-2020-2093.R0)

13-Nov-2020

Dear Mr Furness:

I am writing to inform you that your manuscript RSPB-2020-2093 entitled "Evolutionary simulations clarify and reconcile biodiversity-disturbance models" has, in its current form, been rejected for publication in Proceedings B.

This action has been taken on the advice of referees, who have recommended that substantial revisions are necessary. With this in mind we would be happy to consider a resubmission, provided the comments of the referees are fully addressed. However please note that this is not a provisional acceptance.

Sincerely,
Dr Sasha Dall
mailto: proceedingsb@royalsociety.org

Associate Editor
Comments to Author:

In this paper the authors use an individual-based model to investigate the effect of disturbance environmental heterogeneity on either inhibiting or promoting species diversity. The topic is

clearly interesting to a broad audience, but in its current form I am not sure (as pointed out by one reviewer) that the paper brings enough novel insights that would change our understanding of studied processes. After seen the reviews and reading the paper a second time, my impression was that the interrelationship between the different types of disturbance modeled here (the main results presented in the paper), were, to some extent, a bit trivial/expected. That said there might be more to be explored/presented than what was done in this current version which might lead to new insights. In that respect, as pointed the other reviewer the authors could explore other ecological (or even macroevolutionary) parameters or outcomes of the simulations (e.g. extinction and speciation rates) that could perhaps enrich our understanding of the studied mechanism. In fact, I also felt that further explanation/exploration of some of the simulation results could be interesting, and even better convey the relevance of the simulations done here.

For example, could one extract speciation and extinction rates from those different simulated scenarios? Of course, those do not need to be realistic numbers (the values will depend on the input values for some of the parameters) but showing those values might help one better evaluate what is happening. For example, what explains the difference in species number between different scenarios? Although some of those aspects are mentioned in the discussion (e.g. lines 167-169; or 174-178) I think a graph showing (if possible) those results in the main text could be interesting. My point here is that the approach allows one to further probe what is happening, and this might generate new insights.

Hence, based on the reviewer comments, as well as on my own impressions described above, I unfortunately cannot recommend the paper for publication in its current form. Some of those points might be addressed by further explaining the model but I think it is also important to make it clear how the results presented here advance our understanding of the effect of disturbance in ways that could not be envisioned by previous work or on intuition on how those different kinds of disturbance interact with each other to affect species richness. If the authors think that further explanation/justification or exploration of other aspects of the model would clarify/answer the points raised here, I think a new resubmission would be welcomed.

Reviewer(s)' Comments to Author:

Referee: 1

Comments to the Author(s)

This paper describes a model run of the REvoSim model, where the 100x100 grid is allowed to vary spatially and temporally in different combinations. The overall goal of the paper is to contrast two hypotheses for local diversity, the "patch mosaic" and "climate stability" hypotheses.

The methods as such seem soundly performed, but I'm not convinced that the design of the study actually tests the two hypotheses, nor am I convinced that this model run in general represents a sizable advance in ecological or evolutionary understanding.

First, the two hypotheses seem to not really be alternatives - patch mosaic is to my mind a more ecological hypothesis at recent time scales, whereas I think of environmental stability as a deeper-time evolutionary hypothesis. There is thus no reason to see them as competing. I'm also unsure how the process of combining e.g. pure spatial with spatiotemporal variation - the key approach in the paper - tests these hypotheses. The process of combining itself - which takes the centroid of two numbers in 3d space - does not have a clear ecological / evolutionary interpretation, at least not one explained in the paper.

So in conclusion I find it hard to imagine ecologists reading this paper and significantly changing their view or understanding of these processes. It could possibly still be interesting to publish it in a journal like PLOS ONE.

A very minor detail: it seems a little strange to call a standard linear model with an interaction a "multiplicative" model.

Referee: 2

Comments to the Author(s)

This paper presents an interesting model analysis exploring the potential contribution of disturbance and environmental heterogeneity to biodiversity. The topic is clearly potentially interesting to a broad audience. The paper is generally well written, but it could be made more accessible to a broad audience.

Major comments: I would like to see a clearer communication of the results. The paper's key findings are portrayed in Fig 2, but to the untrained eye the effects of disturbance etc are not easy to ascertain from these plots - which look superficially similar to one another. The Fig legend is also inadequate to explain how these plots communicate the key message around disturbance and biodiversity.

There are numerous ecological parameters included in the models but not extensively explored (competition, dispersal, fecundity, etc). I wonder if comparable results may be obtainable, for instance, from ecologically neutral modelling approaches (e.g. de Aguiar et al 2009 Nature)? To what extent does scale/size of the disturbance interact with the dispersal ability of the organisms?

The MS could also do a clearer job integrating its findings with real-world disturbance scenarios, with more discussion of disturbance-prone systems (compared to the current discussion focus primarily on relatively undisturbed environments such as the deep sea and 'tropics'). E.g. there are some nice examples of ecosystem disturbance and associated biodiversity impacts emerging from studies of tectonically-disturbed intertidal systems (e.g. Parvizi et al 2020 Proc R Soc B).

Minor -

The MS is generally well written, although the first paragraph of the intro becomes almost circular/contorted. Essentially saying: "There are many hypotheses proposed to explain latitudinal biodiversity gradients (sentence 1) because latitudinal diversity gradients can be explained by several different processes (sentence 3)" - sentence 3 seems rather redundant.

Author's Response to Decision Letter for (RSPB-2020-2093.R0)

See Appendix A.

RSPB-2021-0240.R0

Review form: Reviewer 1

Recommendation

Reject - article is not of sufficient interest (we will consider a transfer to another journal)

Scientific importance: Is the manuscript an original and important contribution to its field?

Acceptable

General interest: Is the paper of sufficient general interest?

Acceptable

Quality of the paper: Is the overall quality of the paper suitable?

Good

Is the length of the paper justified?

Yes

Should the paper be seen by a specialist statistical reviewer?

No

Do you have any concerns about statistical analyses in this paper? If so, please specify them explicitly in your report.

No

It is a condition of publication that authors make their supporting data, code and materials available - either as supplementary material or hosted in an external repository. Please rate, if applicable, the supporting data on the following criteria.

Is it accessible?

N/A

Is it clear?

N/A

Is it adequate?

N/A

Do you have any ethical concerns with this paper?

No

Comments to the Author

The last time I reviewed this paper my assessment was that it was solid enough, but not really interesting or conceptually strong enough for Proc B. I also didn't give that many suggestions for changes. As such, it's a little hard for me to tell what I'm supposed to comment on in this re-review.

In general it is an improvement that the authors now explicitly touch upon scale (temporal and spatial) as a means to reunite the two quite distinct hypotheses. It is quite possible in fact that if this were more unfolded there would be a really interesting insight to be gotten from the analysis. I just don't think that it is quite clearly enough unfolded theoretically. The conclusion that both of the mechanisms might contribute, and that this effect is dependent on spatial scale is potentially quite interesting, though, but the discussion relative quickly dives into particulars of this model.

I do acknowledge that the model itself is very nicely built, and the idea of encoding genomes as individual bytes that adapt to RGB space is elegant from a programmer's view point and very computationally efficient. But as I understand it, the model here has been published elsewhere, so this paper should be judged on the merits of the questions addressed here.

My only real technical question that stills need an explanation is how the "blending" is relevant - what kind of situation is the blending supposed to emulate in the different scenarios? I did pose this question in my last review, but it seems like it was misunderstood by the authors.

I think the qualification that it applies to sessile species is quite important for understanding the implications here and would recommend specifying this early, perhaps already in the title.

Line 88: refers to "one principle of evolutionary ecology", which seems like an assertion that would need a reference.

"most previous modelling has been performed at the species level" - is that really true? I've encountered both individual-based and species-based models in my time.

127: each what - grid cell?

Figure 1 shouldn't have a black background imho

Figure 2+3 shouldn't use a diverging color scale, and would be better with a proper categorical scale

"We outline a series of experiments in the methods section" isn't really helpful as a start to the results section

Review form: Reviewer 2

Recommendation

Accept as is

Scientific importance: Is the manuscript an original and important contribution to its field?

Good

General interest: Is the paper of sufficient general interest?

Good

Quality of the paper: Is the overall quality of the paper suitable?

Excellent

Is the length of the paper justified?

Yes

Should the paper be seen by a specialist statistical reviewer?

No

Do you have any concerns about statistical analyses in this paper? If so, please specify them explicitly in your report.

No

It is a condition of publication that authors make their supporting data, code and materials available - either as supplementary material or hosted in an external repository. Please rate, if applicable, the supporting data on the following criteria.

Is it accessible?

Yes

Is it clear?

Yes

Is it adequate?

Yes

Do you have any ethical concerns with this paper?

No

Comments to the Author

This MS has been carefully revised, and overall has been substantially clarified and improved, in line with previous reviewer comments. I support publication in its current form.

Decision letter (RSPB-2021-0240.R0)

19-Mar-2021

Dear Mr Furness

I am pleased to inform you that your manuscript RSPB-2021-0240 entitled "Evolutionary simulations clarify and reconcile biodiversity-disturbance models" has been accepted for publication in Proceedings B.

The referee(s) have recommended publication, but also suggest some minor revisions to your manuscript. Therefore, I invite you to respond to the referee(s)' comments and revise your manuscript. Because the schedule for publication is very tight, it is a condition of publication that you submit the revised version of your manuscript within 7 days. If you do not think you will be able to meet this date please let us know.

Online supplementary material will also carry the title and description provided during submission, so please ensure these are accurate and informative. Note that the Royal Society will not edit or typeset supplementary material and it will be hosted as provided. Please ensure that

the supplementary material includes the paper details (authors, title, journal name, article DOI). Your article DOI will be 10.1098/rspb.[paper ID in form xxxx.xxxx e.g. 10.1098/rspb.2016.0049].

Sincerely,

Dr Sasha Dall

Associate Editor

Board Member

Comments to Author:

I commend the authors for all their hard work to improve the paper. This is a much-improved version of the manuscript. I think many issues have been clarified and it is a lot easier to follow the narrative and the paper's main contributions now. Even though it is almost ready for publication, there are some really minor changes that, if addressed, should improve the quality of the manuscript (see reviewer 1 suggestions). Some are merely aesthetic (e.g. change the black background of figure 1) but other, although small, are not.

Reviewer(s)' Comments to Author:

Referee: 2

Comments to the Author(s).

This MS has been carefully revised, and overall has been substantially clarified and improved, in line with previous reviewer comments. I support publication in its current form.

Referee: 1

Comments to the Author(s).

The last time I reviewed this paper my assessment was that it was solid enough, but not really interesting or conceptually strong enough for Proc B. I also didn't give that many suggestions for changes. As such, it's a little hard for me to tell what I'm supposed to comment on in this re-review.

In general it is an improvement that the authors now explicitly touch upon scale (temporal and spatial) as a means to reunite the two quite distinct hypotheses. It is quite possible in fact that if this were more unfolded there would be a really interesting insight to be gotten from the analysis. I just don't think that it is quite clearly enough unfolded theoretically. The conclusion that both of the mechanisms might contribute, and that this effect is dependent on spatial scale is potentially quite interesting, though, but the discussion relative quickly dives into particulars of this model.

I do acknowledge that the model itself is very nicely built, and the idea of encoding genomes as individual bytes that adapt to RGB space is elegant from a programmer's view point and very computationally efficient. But as I understand it, the model here has been published elsewhere, so this paper should be judged on the merits of the questions addressed here.

My only real technical question that stills need an explanation is how the "blending" is relevant - what kind of situation is the blending supposed to emulate in the different scenarios? I did pose this question in my last review, but it seems like it was misunderstood by the authors.

I think the qualification that it applies to sessile species is quite important for understanding the implications here and would recommend specifying this early, perhaps already in the title.

Line 88: refers to "one principle of evolutionary ecology", which seems like an assertion that would need a reference.

"most previous modelling has been performed at the species level" - is that really true? I've encountered both individual-based and species-based models in my time.

127: each what - grid cell?

Figure 1 shouldn't have a black background imho

Figure 2+3 shouldn't use a diverging color scale, and would be better with a proper categorical scale

"We outline a series of experiments in the methods section" isn't really helpful as a start to the results section

Author's Response to Decision Letter for (RSPB-2021-0240.R0)

See Appendix B.

Decision letter (RSPB-2021-0240.R1)

23-Mar-2021

Dear Mr Furness

I am pleased to inform you that your manuscript entitled "Evolutionary simulations clarify and reconcile biodiversity-disturbance models" has been accepted for publication in Proceedings B.

Your article has been estimated as being 9 pages long. Our Production Office will be able to confirm the exact length at proof stage.

Data Accessibility section

Open Access

Paper charges

Sincerely,

Proceedings B

Appendix A

Dear Editor and Referees

Thank you for your comments. We have broken them down into a series of independent points so that we could assure that they are all addressed. We hope that you find this helpful.

Commenter	Comment	Response
Editor	"I am not sure ... that the paper brings enough novel insights that would change our understanding of studied processes"	To our knowledge, no previous publication has produced a unified framework for the stability-time and patch-mosaic hypotheses of biodiversity. Given that they make opposing predictions regarding the impact of disturbance on species richness, a unified framework is vital to the proper understanding of evolutionary ecology. Furthermore, we have done so using a novel approach, i.e. with a first-principles eco-evolutionary simulation. Our results demonstrate, for the first time, that both mechanisms have previously undocumented limits to their applicability. As Referee 1 points out, the stability-time hypothesis is commonly regarded as something that occurs on evolutionary timescales, whereas the patch-mosaic hypothesis is considered to occur over ecological timescales. The idea that these hypotheses are not in conflict is a misconception based on the ways in which these hypotheses are commonly studied, rather than on the actual eco-evolutionary mechanisms that they entail. Indeed, we demonstrate in this paper that both hypotheses can explain species richness in different scenarios, and they can even act constructively under the right circumstances, but that patch-mosaic disturbance (which can produce diversity on its own) clearly inhibits the stability-time mechanism, if that mechanism would otherwise be strong. Timescale is hence at best misleading, and at worst irrelevant. We consider this an important insight given the implicit misconception that exists in the field at present. We also highlight the importance of spatial scale. Spatial scale has received well-deserved attention in eco-evolutionary circles, but here we demonstrate (to our knowledge, for the first time) that it is spatial scale of disturbance that controls the relative

		importance of the stability-time and patch-mosaic hypotheses. In this sense, we propose the importance of spatial scale as a replacement for the presupposed importance of temporal scale (that we reject). This, too, we consider to be an important insight; it should inform both eco-evolutionary theory and conservation practise in the future. We have revised our manuscript to better communicate the novelty and significance of our results.
Editor	“the interrelationship between the different types of disturbance modelled here ... were, to some extent, a bit trivial/expected”	We agree that the fact that we observe both processes matches our expectations and intuitions, but contend that demonstrating rather than intuiting these relationships is an important advance in and of itself. However we do not consider all of our uncovered relationships to be expected, most especially the strong effect of spatial scale: the stability-time hypothesis typically makes no reference to spatial scale and, as such, our observation that small-scale disturbance can increase species richness would contradict a strict reading of the stability-time hypothesis. The comments of Referee 1 suggest that the ‘expected’ result of our analysis would be that the stability-time hypothesis and patch-mosaic hypothesis do not operate on the same timescale, and are therefore not in conflict. We show that this is not the case, with both occurring on the same timescale and acting destructively or constructively depending on circumstances.
Editor	“the authors could explore other ecological (or even macroevolutionary) parameters”	We feel that this is a request to write a different paper! We absolutely agree that the effects of these ecological parameters are fertile ground for future work, but such an analysis is beyond the scope of this study. We focus on the impacts of disturbance because these present an opportunity to investigate a specific, as-yet unresolved question in evolutionary ecology. The settings for the other parameters are set to levels that are known to show biologically realistic behaviour from previous work (Garwood et al. 2019). While we cannot exclude the possibility that varying other parameters would influence our results, we are confident that our results exist

		in an area of parameter space that is biologically realistic.
Editor	“the authors could explore other ... outcomes of the simulations (e.g. extinction and speciation rates)”	Our original submission did discuss rates of species turnover in the supplementary information (original manuscript supplementary information text and supplementary table 1). This discussion is preliminary, but suggests that disturbance decreases species richness by elevating per-lineage extinction rates rather than by depressing per-lineage speciation rates. A full exploration of these issues is well beyond the scope of the present study, but in response to this suggestion from the editor, we have now moved our existing discussion into the main text, amending the discussion, results, and methods accordingly. We have also expanded our analysis of these factors, as discussed in the following response.
Editor	“For example, what explains the difference in species number between different scenarios? ... I think a graph showing (if possible) those results in the main text could be interesting”	We thank the editor for this suggestion. We had originally considered this issue to be tangential to the main thrust of our paper, but in response to this comment we have undertaken further simulations and analyses to address it and we feel that these significantly strengthen our manuscript. Our new results show that, in the presence of stable niche diversity, equilibrium per-lineage extinction risk is higher in environments that are disturbed, as is predicted by the stability-time hypothesis. However, in the absence of stable niche diversity, equilibrium per-lineage extinction risk is actually reduced by disturbance, which is predicted by the patch-mosaic hypothesis. In both cases, these new data support our original interpretation of the species richness patterns. As suggested, we have added a new figure (Figure 3) to illustrate these results.
Editor	“Some of those points might be addressed by further explaining the model”	We appreciate this suggestion, and hope that the addition of an investigation of origination and extinction rates provides sufficient further explanation of the processes at work in the model. The model itself is detailed in a previous publication: we consider that reproducing those details here would be unnecessary repetition, but if the editor would prefer them to be recapitulated in supplementary information we can of course

		do so. The REvoSim model is a reasonable, if simplified, simulation of real ecosystem processes at the most fundamental possible level. We believe that this makes criticism of the model on the grounds of inaccuracy difficult, as the assumptions are few and well supported. Consequently, processes produced by the model are very likely to occur in the real world, and its reproduction of realistic patterns (in both the original publication and here) support this assertion.
Editor	“I think it is also important to make it clear how the results presented here advance our understanding of the effect of disturbance in ways that could not be envisioned by previous work or on intuition on how those different kinds of disturbance interact with each other to affect species richness”	Because they are often considered on different timescales there has not, to our knowledge, been a previous attempt to reconcile the stability-time and patch-mosaic hypotheses. Our study demonstrates that, contrary to the assumptions of many workers, including Referee 1, timescale is not the key difference between these processes. Both influence long term species richness, and the mechanisms do conflict in practice, as we show: the same disturbance that creates patch-mosaic effects also inhibits stability-time effects. Rather, the key difference is spatial scale of disturbance relative to scale of background spatial heterogeneity. This importance of spatial scale is a novel finding. This was not clear enough in our original submission. As such we have adjusted the introduction and discussion to highlight the novel and impactful areas of our results, specifically the importance of spatial scale differences and the unimportance (or, at least, unnecessary) of temporal scale differences.
Referee 1	“the two hypotheses seem to not really be alternatives - patch mosaic is to my mind a more ecological hypothesis at recent time scales, whereas I think of environmental stability as a deeper-time evolutionary hypothesis.”	We especially thank the referee for expressing this concern, because this is a view that we seek to refute in this paper; our first version failed to make this clear enough. We have inserted a new paragraph into the introduction to clarify this argument. We hope that the statement, now in the main text, “organisms that are only present in ecosystems because of disturbance on ecological timescales ... could not have evolved, and cannot persist as species, without a steady supply of disturbed ecosystems being produced over evolutionary time”, helps to better explain our argument. We would argue that one principle of

		evolutionary ecology is that the partition of phenomena into evolutionary and ecological timescales should not be an a priori assumption.
Referee 1	"I'm also unsure how the process of combining e.g. pure spatial with spatiotemporal variation - the key approach in the paper - tests these hypotheses"	We assume that this reflects a lack of clarity in our explanation of the link between ecological theory and our experimental methods, rather than an issue with the experimental design itself. We have added a more substantial explanation at the end of the introduction to clarify this. The combination of these factors is a necessary component of the study because environments will exist in nature with a combination of these factors acting simultaneously. The magnitude and polarity of the effect of one factor on species richness is liable to be contingent upon the presence or absence of the other factors. Our results support this. We have significant interaction terms in our linear models, which we can only detect because our experimental design involved the use of combinations of environment types.
Referee 1	"The process of combining itself - which takes the centroid of two numbers in 3d space - does not have a clear ecological / evolutionary interpretation, at least not one explained in the paper"	We have added a further explanation and clarification of our rationale to the start and end of the Environment Generation section of the methods.
Referee 1	"it seems a little strange to call a standard linear model with an interaction a "multiplicative" model."	Our phraseology was indeed confusing, sacrificing precision for conciseness. We have changed all references to multiplicative models to standard linear models with interaction terms.
Referee 2	"I would like to see a clearer communication of the results. The paper's key findings are portrayed in Fig 2, but to the untrained eye the effects of disturbance etc are not easy to ascertain from these plots - which look superficially similar to one another."	We thank the referee for this feedback. Figure 2 has been updated with a new set of subfigures, where the species richness axis has been log transformed in order to show the biologically important difference in species richness between low PS – high PT and low PS – high ST conditions. The difficulties in displaying the shape of the graph surface that this transformation presents have been circumvented by presenting a 2D rather than 3D visualisation, viewed from above with a colour gradient

		delineating species richness (rather than a 3 rd vertical axis).
Referee 2	“The Fig legend is also inadequate to explain how these plots communicate the key message around disturbance and biodiversity.”	We have added additional information to the figure legend to highlight the way in which the plots support our key message.
Referee 2	“There are numerous ecological parameters included in the models but not extensively explored (competition, dispersal, fecundity, etc).”	See response to a similar comment by the Editor; we agree that this would be interesting, but consider it beyond the scope of this paper.
Referee 2	“I wonder if comparable results may be obtainable, for instance, from ecologically neutral modelling approaches (e.g. de Aguiar et al 2009 Nature)?”	We thank the referee for drawing our attention to this paper. The idea of replicating our study using neutral models is interesting because it highlights the dual nature of disturbance in ecosystems: disturbance can independently modify the fitness landscape of an area and directly remove individuals from it. While both effects are presumably present in our experiments, our study focusses on the former because it is this effect that both the stability-time and patch-mosaic hypotheses are concerned with. By contrast, neutral models exclusively consider the latter, as modification of the fitness landscape is not a meaningful concept in a neutral model, where the fitness landscape must, by definition, be flat. If neutral models could replicate the results of our study using only the latter form of disturbance, where individuals are removed from the ecosystem, then this would cast doubt on our claim that the stability-time and area-mosaic hypotheses are the driving forces behind our observed changes in species richness. However, we are confident that neutral models could not achieve this for two reasons: firstly, in a neutral model, an ecosystem with no temporal disturbance is identical to another ecosystem with no temporal disturbance, regardless of the structure of the undisturbed ecosystem. Consequently, in such a model, we would have expected to observe no difference in species richness between low and high PS environments, which we do. Secondly, a neutral model would make very little distinction between ST and PT disturbances, where a non-neutral model, such as ours,

		views them as very different. A neutral model could therefore not reproduce the very different effects that these two forms of disturbance have on species richness in our model. We have added discussion on this matter to our discussion section
Referee 2	“To what extent does scale/size of the disturbance interact with the dispersal ability of the organisms?”	We agree with the referee that it would be interesting to investigate how different dispersal abilities influence the results of our simulations but, as the referee has previously pointed out, there are a large number of parameters in our simulation that we have the ability to vary, and the volume of data required to study them properly increases exponentially with the number of parameters investigated. Consequently, we consider this also to be beyond the scope of this study, but of interest for future studies.
Referee 2	“The MS could also do a clearer job integrating its findings with real-world disturbance scenarios, with more discussion of disturbance-prone systems (compared to the current discussion focus primarily on relatively undisturbed environments such as the deep sea and 'tropics'). E.g. there are some nice examples of ecosystem disturbance and associated biodiversity impacts emerging from studies of tectonically-disturbed intertidal systems (e.g. Parvizi et al 2020 Proc R Soc B).”	We thank the referee for calling our attention to this reference: we have incorporated it into the text and it, along with their previously mentioned reference to de Aguiar et al. (2009), helped us to improve the discussion on the dual nature of disturbance. The founder effects observed in Parivizi et al. (2020) are not related to the patch-mosaic hypothesis, but they do represent a plausible alternative hypothesis for why high disturbance ecosystems might have high species richness. As such, we have added to the discussion a paragraph on neutral processes such as founder effects, and an explanation of why they are insufficient to explain our results. With respect to the discussion of disturbance prone vs undisturbed environments, it is worth noting that we do not suggest that the deep sea is relatively undisturbed, merely relatively unaffected by large scale disturbance. Indeed, we join Grassle and Sanders (1973), and others, in ascribing deep-sea diversity to small-scale disturbance events. Nevertheless, we agree with the referee that the original manuscript lacked an example of an area with high levels of large-scale disturbance; we have added a brief discussion of previously glaciated areas of the

		globe, the relatively low biodiversity of which can be explained by our results.
Referee 2	“the MS is generally well written, although the first paragraph of the intro becomes almost circular/contorted.”	This paragraph has been re-written to address this (all too true) criticism.

Appendix B

Dear Editor and Referees

Thank you for your comments. We have broken them down into a series of independent points so that we could assure that they are all addressed. We hope that you find this helpful.

Commenter	Comment	Response
Referee 2	I support publication in its current form.	No changes required on the basis of this comment.
Referee 1	My only real technical question that stills need an explanation is how the "blending" is relevant - what kind of situation is the blending supposed to emulate in the different scenarios? I did pose this question in my last review, but it seems like it was misunderstood by the authors.	Neither we nor reviewers have previously used the term "blending". We assume, therefore, that this refers to the process of combining two heterogeneity types in the production of environment files. Real world environments typically contain heterogeneity on multiple different spatial and temporal scales (e.g. climate change vs topography vs landslides), and we considered it likely that the impact of one form of heterogeneity on species richness would be dependent on the presence or absence of other forms. Our results confirm this, and we could only obtain those results by combining multiple types of disturbance within environment sets. As for what they represent, PS represents long-term spatial heterogeneity (e.g. topography), PT represents large scale changes (e.g. climate patterns), and PS represents small scale changes (e.g. landslides, or tree falls). We have added a note to this effect at the end of the environment generation section of the methods.
Referee 1	I think the qualification that it applies to sessile species is quite important for understanding the implications here and would recommend specifying this early, perhaps already in the title	We do not wish to over-emphasise this point as the organisms are quite capable of dispersal during reproduction and, from a genetic point of view, there is not a great difference between sessile and mobile taxa. We have added a note to the abstract clarifying that these organisms are sessile, and the introduction repeats this point.

Referee 1	Line 88: refers to "one principle of evolutionary ecology", which seems like an assertion that would need a reference.	Reference added.
Referee 1	"most previous modelling has been performed at the species level" - is that really true? I've encountered both individual-based and species-based models in my time.	We believe that this is true. Individual-level models do exist (e.g. Avida), but modelling long timescale evolution at the individual level has historically been hindered by computing requirements. This is one of the great draws of REvoSim as a tool: it is computationally efficient. We have modified the statement in the manuscript to ensure that it is not misleading.
Referee 1	127: each what - grid cell?	Correct. Clarification added.
Referee 1	Figure 1 shouldn't have a black background imho	We have remade this figure with a white background.
Referee 1	Figure 2+3 shouldn't use a diverging color scale, and would be better with a proper categorical scale	We do not understand this comment: a categorical colour scale consists of multiple, unrelated colours and, as the name suggests, is suited to categorical, not continuous, data. Our data are continuous. We have not made a change to the manuscript on the basis of this comment.
Referee 1	"We outline a series of experiments in the methods section" isn't really helpful as a start to the results section	Sentence removed.